



# A technical description of the Balloon Lidar Experiment BOLIDE

Bernd Kaifler[1], Dimitry Rempel[1], Philipp Roßi[1], Christian Büdenbender[1], Natalie Kaifler[1], and
Volodymyr Baturkin[2]

[1]Deutsches Zentrum für Luft- und Raumfahrt, Institut für Physik der Atmosphäre, Oberpfaffenhofen, Germany
[2]Deutsches Zentrum für Luft- und Raumfahrt, Institut für Raumfahrtsysteme, Bremen Germany

**Correspondence:** Bernd Kaifler (bernd.kaifler@dlr.de)

**Abstract.** The Balloon Lidar Experiment (BOLIDE) was the first high-power lidar flown and operated successfully onboard
a balloon platform. As part of the PMC Turbo payload, the instrument acquired high resolution backscatter profiles of Polar
Mesospheric Clouds (PMCs) from an altitude of ∼38 km during its maiden ∼6 day flight from Esrange, Sweden, to Northern
Canada in July 2018. We describe the BOLIDE instrument and its development and report on the predicted and actual in-flight
performance. Although the instrument suffered from excessively high background noise, we were able to detect PMCs with a
volume backscatter coefficient as low as $0.6 \times 10^{-10} \mathrm{m}^{-1} \mathrm{sr}^{-1}$ at a vertical resolution of $100\,\mathrm{m}$ and a time resolution of $30\,\mathrm{s}$.

## 1 Introduction

Since several decades light detection and ranging (LIDAR[1]) is still the only technology which allows for profiling of the
neutral atmosphere from the troposphere to the upper mesosphere and lower thermosphere. Although the basic principle has
not changed since the first experiments using powerful searchlights as light source, it was the invention of the laser which
facilitated the technological breakthrough and adoption of the lidar method for atmospheric research. First observations of
tropospheric clouds by lidar were reported in the 1960s and were soon followed by observations of stratospheric aerosols (e.g.
Collis, 1965, 1966; Schuster, 1970). However, it took another decade before the technology became mature enough to allow
for measurements of air densities with sufficient precision and accuracy for the retrieval of mesospheric temperature profiles
by hydrostatic integration (Hauchecorne and Chanin, 1980).

The push to ever greater heights and better time and altitude resolutions was mainly hindered by two things: First, the density
of air decreases exponentially with altitude, causing the lidar return signal to decrease exponentially with altitude too. Second,
in the picture of geometric optics the scattered light forms spherical waves, resulting in the lidar return signal decreasing
with the range squared (or altitude squared in case of a vertically looking lidar). Both factors are responsible for the rapid
deterioration of the received signal and put a natural limit on the altitude where the wanted signal fades into background.
One pragmatic although from a technical point of view less straight forward answer to this problem was power scaling of
lidar systems i.e. the use of more powerful lasers, larger telescopes, and more sensitive detectors (von Zahn et al., 2000;
Wickwar et al., 2016). Another more creative solution was first demonstrated in the late 1970s. Gibson et al. (1979) showed
that mesospheric temperature could be retrieved from probing of the Doppler-broadened sodium $D_1$ and $D_2$ lines. Although the

---

[1]Instead of the acronym we will use the term *lidar* from now on.


sodium density within the metal layer (approximately 80-100 km altitude) is only in the order of few thousand atoms per cubic centimeter and thus much smaller than the air density at these altitudes, the sodium fluorescence signal is nevertheless much stronger than Rayleigh scattering due to the ~16 order of magnitude difference in the scattering cross-sections. The downside of these so-called metal fluorescence (or resonance) lidars was the technical complexity of the lasers. While Rayleigh scattering is observed for any wavelength, fluorescence lidars require precise tuning of the laser wavelength to the absorption line of the

probed metal atoms. This requirement resulted in bulky and less reliable lasers which were often cumbersome to operate. Hence, metal fluorescence lidars became never as widely used as Rayleigh lidars.

Another option to boost the lidar return signal is to decrease the distance between the lidar and the probed volume. As mentioned before, the lidar return signal decreases with the distance squared when disregarding the exponential decrease in air density. Thus, for example, a lidar flying at 40 km altitude and probing a volume at 80 km experiences a fourfold increase of

the wanted signal over a ground-based lidar probing the same volume. Furthermore, with more than 99% of the air below, the upward looking flying lidar collects much less scattered sunlight than its ground-based counterpart. The low solar background would permit the flying lidar to make observations in full daylight without the ultra-narrowband optical filters required in receivers of ground-based lidars. Omitting the filters does not only considerably simplify the optical design of the lidar, but at the same time increases the overall transmission of the instrument. The latter facilitates the use of even smaller lasers

and/or telescopes while still achieving the same signal level as large ground-based instruments. Taking into account above considerations, the concept of a compact but powerful balloon-borne mesospheric lidar appears feasible and enticing. Greatest benefits from such an instrument are expected for studies which require observations in daylight e.g. high-resolution profiling of polar mesospheric clouds (PMC).

With the mass and power requirements met (typically few hundred kilograms and up to 1 kW for large balloon payloads),

the technological challenges of getting a lidar to work in the near-space environment are still formidable in comparison to ground-based systems. The balloon experiment has to withstand low pressure, intense solar radiation, and extreme temperatures ranging from -60°C encountered during ascent and descent to >100°C in sunlit conditions at floating altitude. With not enough air pressure for convective cooling, the only means to dump waste heat generated by the instrument is to radiate it into space. Any instrument which dissipates a significant amount of electrical power requires a radiator for cooling and thermal shielding

similar to satellites. Furthermore, on long duration balloon flights communication between ground station and balloon payload typically relies on satellite links which may not be available all the time and are severely limited in bandwidth. Thus, the experiment is required to have the capability to run autonomously for at least some periods and needs to incorporate fault protection routines to deal with potential anomalies. Hence, the design of a balloon lidar system faces challenges which are very similar to satellite instruments. While mass, volume and power requirements maybe less restrictive for a balloon instrument,

thermal control is actually more challenging. In addition to radiative heating and cooling experienced in (near-)space, the balloon instrument has to cope with a variable amount of (forced) convective cooling, in particular during the ascent and descend of the balloon through the cold tropopause. One of the biggest differences, also from a management point of view, is that balloon projects typically operate on a much smaller budget and compressed schedule.




| Parameter | Requirement | Performance |
|---|---|---|
| Mass | <150 kg | 143 kg |
| Maximum height | 1.6 m | 1.42 m |
| Power | <700 W | 630 W |
| Viewing direction | >26° off zenith | 28° off zenith |
| Range resolution | 10 m | 3 m |
| Time resolution | 10 s | 1 s |
| PMC detection threshold | $1 \times 10^{-10} \mathrm{m}^{-1}\mathrm{sr}^{-1}$ | $0.58 \times 10^{-10} \mathrm{m}^{-1}\mathrm{sr}^{-1}$ |
| Temperature retrieval | 50–75 km | 50–78 km |

**Table 1.** BOLIDE instrument requirements and actual performance. The PMC detection threshold is defined for 30 s time resolution and 100 m vertical resolution.

Roots of the Balloon Lidar Experiment (BOLIDE) project go back to the 12th International Workshop on Layered Phe-
nomena in the Mesopause Region (LPMR) held in Boulder, USA in 2015. D. Fritts (GATS Inc., Boulder Division) gave a
presentation on a recently selected NASA long duration balloon mission carrying a suite of narrow and wide field cameras for
imaging of polar mesospheric clouds (PMCs). The mission was called PMC Turbo and the payload was to be launched from
McMurdo in Antarctica in the 2017/2018 season on a flight lasting approximately 14 days (Fritts et al., 2019). Also present
during the workshop was B. Kaifler (DLR) who proposed to augment the payload with a lidar for near vertical profiling and
altimetry of PMCs, information which cannot be retrieved from the planned imaging experiments, but is critical for the inter-
pretation of the acquired images. The proposal lead to an agreement between NASA and DLR in January 2016, when NASA
formally accepted the balloon lidar experiment as contributed instrument.

In addition to the science objectives, the BOLIDE project served also as "proof-of-concept" for various technologies e.g.
operation of an efficient single-loop liquid cooling system on a balloon platform. The instrument was also remarkable for its
rapid development cycle and its low cost relative to other balloon instruments of similar complexity. In this regard BOLIDE
will also hopefully serve as pathfinder for future balloon instruments much like the *Mars Pathfinder* mission did for robotic
space missions (Golombek et al., 1999). Before BOLIDE there was only one previous attempt to fly a high-power lidar on
a balloon. The *Balloon Winds* payload, a technology demonstrator for a future satellite mission, was, however, lost during a
launch mishap and no in-flight data were collected (Dehring et al., 2006).

## 2 Requirements and early design phase

In the early conceptual design phase following the LPMR workshop requirements were defined as follows. 1) The lidar is to
provide PMC backscatter profiles with higher resolution and similar PMC detection threshold as the ground-based ALOMAR
RMR-lidar i.e. 30 s temporal and 100 m vertical resolution with a detection threshold of $\Delta\beta_{\mathrm{PMC}} = 1 \times 10^{-10} \mathrm{m}^{-1}\mathrm{sr}^{-1}$ (for
comparison: Fiedler et al. (2011) used 14 min, 150 m, and $\Delta\beta_{\mathrm{PMC}} = 1 \times 10^{-10} \mathrm{m}^{-1}\mathrm{sr}^{-1}$ for processing ALOMAR data). 2)





The lidar is to provide temperature profiles below the PMC layer in the range ∼50–75 km. 3) The average power consumption is <700 W and the mass should not exceed 150 kg, and 4) the height of the telescope is limited to 1.6 m. The latter resulted from the maximum permissible height of the gondola and the constraint to place the telescope in the shadow zone of the gondola. From the very beginning PMC Turbo was designed as pointed payload, using an azimuth rotator to track the sun and turn the science instruments to anti-solar direction. With the solar array mounted on the sun-facing side of the gondola and

science instruments placed on the opposite side, a permanent shadow zone is created behind the solar array which protects the instruments from direct solar radiation (see Figure 4). 5) The viewing direction is limited to >26 degrees off-zenith. This requirement follows from the size of the balloon above the gondola and the length of the flight train, as for obvious reasons the laser beam is not allowed to hit the balloon. The requirements are summarized in Table 1.

      For the early design studies DLR's mobile ground-based lidar systems served as starting point for performance studies as

well as for weight and power calculations (e.g. Kaifler et al., 2015). These lidars use diode pumped solid state lasers with 12 W output power at 532 nm wavelength as light source and a fiber coupled, 24 inch diameter, f/2.4 mirror as receiving telescope. While the optical power of the laser was certainly sufficient, the electrical power consumption and a projected mass of ∼40 kg for the laser alone became a problem. The real show-stopper was, however, the need to pump deionized water through the laser pump chambers for cooling. Not only would the use of water as primary coolant require a dual loop cooling system,

as water in the external radiator is likely to freeze during the ascent through the cold tropopause, but it would also result in guaranteed destruction of the laser upon landing. Recovery of the payload on the Antarctic plateau is estimated to take a few days at best, and during that time the payload is subject to the cold Antarctic environment with the laser eventually reaching sub-zero temperatures. These problems lead to selection of a less powerful but passively cooled laser which could be mounted on a cooling plate instead of running coolant directly through the pump chambers.

Further significant design changes included the selection of a smaller telescope to satisfy the height criterion and the reduction of the telescope field of view (FOV) to 180 μrad. The latter was motivated by the corresponding reduction in the solar background which is proportional to the FOV squared. The requirement for ground tests in daylight followed from selection of the launch site. Because the lidar would be disassembled before shipment to McMurdo, Antarctica and deployment was scheduled to take place in November when the sun is already up 24 h a day, tests in daylight were seen as the only means to

carry out a full system test before the actual balloon launch. The use of a narrow-band etalon in the receiver for filtering out of the solar background was dismissed early on because this would have required a narrow-band transmitter i.e. a seeded laser. Since the BOLIDE instrument was regarded as the first test of a mesospheric lidar system on a balloon, it was designed for simplicity and reliability and thus lacked a seeded laser. Without any possibility to sufficiently filter out the solar background, testing the lidar in daylight could only be achieved by inserting neutral density filters into the optical path of the receiver. That

obviously also attenuates the wanted signal, but reduces the total signal, i.e. lidar return signal plus solar background, to levels manageable by the sensitive detectors. This approach was deemed acceptable because the operation at ground in daylight was required for engineering tests only.





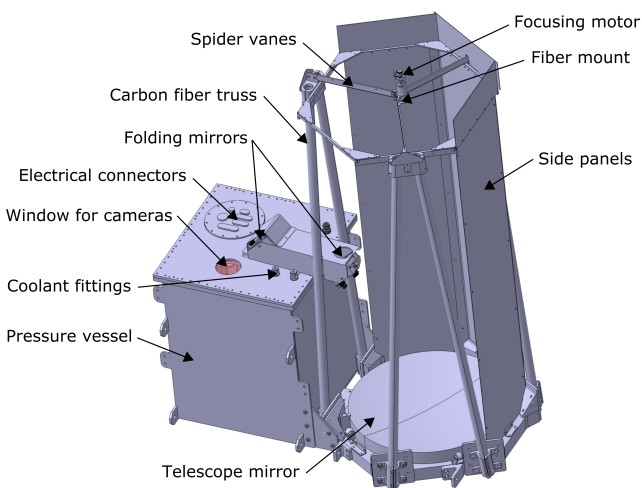

**Figure 1.** Computer rendering of the pressure vessel with the telescope attached (3 side panels removed).

With these critical design decisions made very early in the project, the design of what became the BOLIDE instrument was completed by late summer 2016. With the original launch date less than 1.5 years away, the implementation phase had to follow

immediately. In particular, there was no time for testing the design on a breadboard or brassboard level.

A description of the PMC Turbo gondola and its power and communication systems can be found in Kjellstrand et al. (submitted).

## 3 Description of the flight instrument

The flight configuration of the BOLIDE instrument comprises three main components: 1) the pressure vessel housing the laser

transmitter, receiver, electronics, main computer and cooling system, 2) the receiving telescope, and 3) the radiator. A rendering of the instrument is shown in Figure 1.

### 3.1 Optical setup

Before going into details of the mechanical construction, let us first have a look at the optical setup depicted in Figure 2. The laser is a frequency-doubled master oscillator power amplifier (MOPA) system built by Montfort Laser GmbH with 4.2 W

average output power at 532 nm wavelength and 100 Hz pulse repetition frequency and 5 ns pulse length. The laser uses thermoelectric coolers for heat removal and thermal stabilization and can thus be fully conductively cooled with no requirement for circulating coolant directly through the pump chambers. In case of BOLIDE, the laser is mounted on an aluminum plate which serves as both structural member and heat sink. Glycol is pumped through channels embedded in the plate to remove waste heat. Not converted 1064 nm light is separated by a dichroic mirror and subsequently dumped. A turning mirror mounted

on a fast piezo driven tip-tilt platform with 3.5 mrad angular range directs the remaining 532 nm beam to the first beam expander





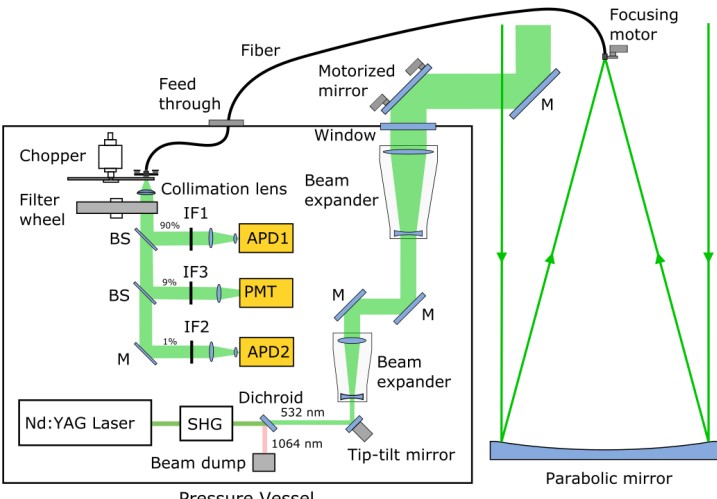

**Figure 2.** Schematics of the optical system; APD = avalanche photodiode, BS = beam splitter, IF = interference filter, M = turning mirror, PMT = photomultiplier tube, SHG = second harmonic generator. See text for details.

(4x). This beam expander is motorized for in-flight adjustment of the beam divergence. Further turning mirrors guide the beam to the second beam expander which increases the beam diameter 2.5-fold to approximately 40 mm and reduces the beam divergence to 77 μrad and 56 μrad for the fast and slow axis. The expanded beam exists the pressure vessel through a 102 mm diameter anti-reflective coated window. A final set of beam turning mirrors relay the beam into the tube of the receiving telescope where it is transmitted into the sky at a location 0.21 m offset from the optical axis of the receiving telescope. Piezo drives on the first mirror allow for coarse adjustments to the beam pointing, while the piezo driven tip-tilt mirror is used to fine control the overlap between laser beam and FOV of the telescope.

The receiving telescope uses a 0.5 m diameter f/2.4 quartz mirror with 38 mm edge thickness supported by a three-armed mirror cell. Light collected by the mirror is coupled directly into an Optran UV fiber (200 μm core diameter, NA 0.22) mounted in the focal plane, and a motorized spring-loaded vertical translation stage allows for fine-tuning of the telescope focus in flight. The resulting FOV of the telescope as determined by the diameter of the fiber core is 165 μrad. The spot size of the mirror (15 μm) is much smaller than the facet of the fiber, thus coupling losses are negligible except for 8% losses due to uncoated fiber ends. In order to minimize focus changes related to variations in temperature, the central spider holding the fiber mount is supported by a three-legged truss made of carbon fiber tubes.

The optical fiber enters the lidar pressure vessel through a vacuum feed through and terminates in front of a mechanical chopper rotating at 100 rotations per second synchronized to laser firing times. The chopper blocks the intense signal originating from scattering of the laser beam in the vicinity of the receiving telescope, thus preventing the detectors from overloading. A collimation lens collimates the beam before passing through a filter wheel loaded with neutral density filters with exception of one clear position. The filters are used only during ground tests and allow for sufficient attenuation of the solar background



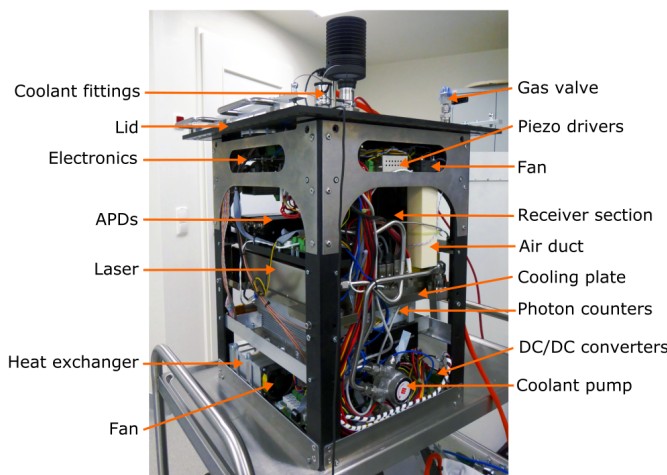

**Figure 3.** The pressurized section containing the lidar receiver, laser and electronics. This picture was taking during integration with the bottom half of the pressure vessel missing.

for operation of the detectors in daylight. In flight, the clear position is used exclusively. Finally, as shown in Figure 2, the collimated beam is splitted into three beams which are directed to individual detectors: two avalanche photo diodes operated in Geiger mode (APD1 and APD2: Excelitas SPCM-ARQH-16) and a photomultiplier tube (PMT: Hamamatsu H12386-210). The splitting ratio between the three channels is approximately 90:9:1 with 90% of the light reaching the upper detector APD1. Both avalanche photo diodes (APD1 and APD2) are gated such that peak photon counting rates remain below 7 MHz in order

to limit signal induced noise which results from significant heating of the silicon chip. Narrow band interference filters in front of the detectors reduce the solar background and minimize cross-talk between detectors. Filters used are: 0.3 nm bandwidth (FWHM) and 70% peak transmission (IF1), 0.8 nm bandwidth and 81% peak transmission (IF2), and 1 nm bandwidth and 50% peak transmission (IF3). The electrical pulses coming from the detectors are time stamped with 800 ps precision corresponding to 0.12 m range resolution using a three-channel MCS6A multiscaler from FastCOMTec GmbH.

**3.2 Mechanical setup**

In order to keep the costs of the BOLIDE instrument low, the decision was made to use off the shelf components wherever possible. This included the use of non-vacuum compatible components. Hence, the instrument was to be housed in a pressure vessel designed to withstand 1 bar differential pressure when sealed on ground. The vessel is a two piece aluminum structure with a welded, cube shaped lower half of size 0.44 x 0.44 x 0.57 $m^3$ and a machined lid. The lower part comprises 5 mm thick

plates, a 10 mm thick flange, and welded-on straps which represent the structural interface to the balloon gondola. The lid contains the optical windows, feed throughs for the optical fiber, coolant lines, and electrical cables, as well as a relief valve for pressure equalization. Lid and lower part are held together by 64 bolts and are sealed off using a single O-ring.





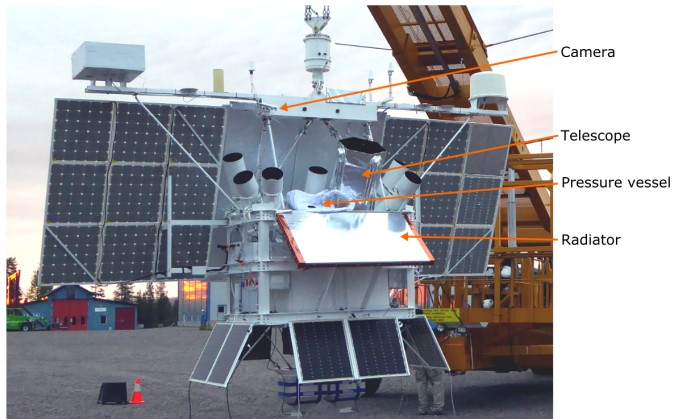

**Figure 4.** The PMC Turbo gondola during launch preparations on 6 July 2018.

The internal parts of BOLIDE are distributed over five shelves mounted in a rack which is bolted to the lid. Having a firm connection between rack and lid rather between rack and bottom half simplifies considerably the integration, as all optical, electrical and fluid connections run through the lid. The shelves are stacked during integration from bottom to top and comprise following sections: the cooling section, the high speed photon counter, the laser mounted on the cooling plate, the receiver, and the electronics section. The cooling section houses a welded glycol coolant tank with 6 l volume, the coolant pump, and DC/DC converters for the laser, pump, and the 12 V main bus. All components are mounted on a common aluminum plate which also serves as the bottom of the glycol tank and thus allows for good conduction of waste heat into the coolant. A heat sink and fan assembly which is thermally bonded to one of the walls of the tank removes waste heat from the air inside the pressure vessel by forced convection.

The second shelf carries the photon counting electronics whose heat sink is thermally contacted to the lower face of the laser cooling plate. Next follows the laser with the second beam expander protruding vertically into the receiver section above. The receiver shelf carries the optical assembly of the receiver including chopper, filter wheel and detectors. It carries also the mount for the infrared airglow camera contributed by Utah State University. The electronics shelf carries electronics modules on both sides and serves also as heat sink for the devices. Main components are the main computer, the FPGA board, piezo drivers, microcontrollers, power switches, and signal converters.

### 3.3 Liquid cooling system

The active cooling system uses a single glycol cooling loop to remove waste heat generated inside the pressure vessel. A rotary vane pump draws coolant (brand name Tyfocor Art LS) from the glycol tank which acts as both reservoir and accumulator and is located at the lowermost shelf. The liquid passes then through the laser cooling plate and the external radiator, where the waste heat is finally radiated into space, before it is returned to the tank at a lower temperature. The one-sided radiator has an active surface of $1.6 \, \text{m}^2$ and is capable of rejecting 550 W at $18°\text{C}$ inlet temperature and flow rate of $6 \, \text{l} \, \text{min}^{-1}$. Electrical bias



heaters mounted at the back side of the radiator are used to control the temperature of the glycol at the outlet and thus inside

the glycol tank. The operating temperature of the cooling loop is constrained to 16–22°C, which is the temperature range the

laser system was tested. In order to keep the heat load on the radiator and thus the glycol temperature constant, replacement

heaters located at the bottom of the glycol tank are activated when the laser is not in use. A detailed description of the thermal

control system and analysis of its performance will be published in a separate study.

### 3.4    Power system

The BOLIDE instrument is supplied with unregulated 28 VDC primary power from the battery bus of the gondola. The

primary power is internally distributed to a 12 V DC/DC converter, which supplies the main instrument bus, and two 24 V

DC/DC converter providing power to the laser and the coolant pump. Moreover, the laser replacement heater and radiator

bias heaters are connected to the primary power via computer controlled relays. All other electrical equipment is supplied

from the 12 V instrument bus or a 5 V secondary bus. Each electrical system can be isolated through a series of power

switches which are controlled by the main computer. Voltage and current sensors in the four secondary busses and the primary

bus allow monitoring of the power consumption and voltage levels. The information is used by fault protection algorithms

implemented in the main computer to assess the health of the instrument, as well as to protect the instrument from undervoltage

conditions and overloads. All power switches are configured as normally open, causing the switches to open and automatically

isolate downstream equipment should an undervoltage condition occur. The only exception is the switch through which power

is supplied to the main computer. This switch is controlled by a dedicated microcontroller and is configured as normally

closed in order to allow for automatic power up of the computer when power is supplied to the instrument. A watchdog timer

implemented in the microcontroller cycles the power switch should the software running on the main computer lock up and

fail to send a heartbeat signal to the microcontroller. Power cycling causes an automatic restart of the computer followed by

entry into safe mode with all equipment not critical for survival of the instrument being powered off.

### 3.5    Computer systems

BOLIDE uses single board computer (SBC) produced by Advantech as main computer. The MIO-5271Z-4GA9A1E SBC

features an Intel i5-4300U CPU running at 1.9 GHz clock rate and 4 GB RAM, and consumes about 15 W power under load.

The SBCs two Ethernet ports are connected to the local area network of the gondola for air to ground communication and to the

embedded controller/FPGA which acts primarily as a multiplexer/demultiplexer for the various sensors, actuators and power

switches used in BOLIDE. The photon counter and cameras are connected to the computer via USB, while the laser controller

and microcontroller use RS-232 interfaces. For program and data storage, the computer uses three redundant solid state disks

with a total capacity of 4 TB. The computer runs Linux as operating system and custom application software written in C/C++.

The embedded controller/FPGA board is a sbRIO-9637 with a 667 MHz dual-core processor and a Zynq-720 field pro-

grammable gate array (FPGA) from National Instruments. The primary intended use of the FPGA is the implementation of

interfaces to sensor and control busses and the implementation of the synchronization logic and timing loops which control all

time critical tasks e.g. triggering of laser pulses, start of data acquisition, and gating of detectors. Furthermore, several safety





features with regard to operation of the laser are implemented in the FPGA. These include the automatic shutdown of the laser in case of an out of bounds attitude of the gondola, out of bounds altitude, or when the command loss timer expires.

### 3.6 Flight Data System and Command & Control System

The Flight Data System (FDS) comprises software running on both the main computer and the embedded controller/FPGA. Its main purpose is to collect science data from the photon counter and cameras as well as telemetry from various subsystems, and to format the data for transmission to ground and for onboard storage. Two engineering and 7 mixed science and engineering data formats which require downlink bit rates ranging between $1.2$ kbit/s and $26$ kbit/s are implemented in the FDS. A basic science frame containing a full photon count profile of the high rate channel (detector APD1, see figure 2) with $100$ m 230 range resolution is included in all mixed formats and sent to ground every two seconds. These high priority science data are interleaved with basic engineering data and a variable number of lower priority sub frames filled with higher resolution science data. All frames are compressed using zlib before being queued for downlink. While the high priority frames are retrieved from the queue and sent to ground immediately, the low priority frames remain in the queue until a frame fits into the downlink data stream whose capacity is defined by the selected bit rate. This scheme maximizes the amount of science data reaching the 235 ground while taking into account the variable efficiency of data compression.

The Command & Control System (CCS) receives commands from the ground, validates the commands, and stores the commands in a sequencer table. A software loop scans the table 10 times per second for commands which are tagged for execution. These commands are then removed from the table and forwarded to the appropriate subsystems.

Commands are sent from ground in command loads which are lists of commands encapsulated in special headers and footers. 240 The latter are used by the CCS for validation of a command load and verifying its integrity via check sums. Commands can be either real-time commands which are executed as soon as they are received by the CCS, or time-tagged commands which are scheduled by the sequencer for execution at later times. With time-tagged commands, execution times can be specified as absolute times, i.e. relative to the epoch of the main clock, or as time offsets relative to the execution time of a previous command. This sequencing capability allows for a great deal of flexibility in controlling of the instrument even when no 245 satellite link is available for real-time commanding. One in particular critical part of the mission without real-time commanding capability is the descent phase when the thermal control system needs to be reconfigured as a result of the rapidly changing environmental conditions.

### 4 Predicted and in-flight performance

The performance of the lidar instrument was predicted by solving the lidar equation for the given set of instrument parameters 250 (see Table 2). In order to model the solar background picked up by the instrument, sky radiances were calculated using the libRadtran software packet for radiative transfer calculations (Emde et al., 2016) and converted to photon counting rates. These background rates were then integrated taking into account the length of the lidar range gates and number of laser pulses and





| Laser average optical power | 4.2 W |
|---|---|
| Laser pulse rate | 100 Hz |
| Telescope/laser bore sight efficiency | 1.0 |
| Telescope diameter | 0.5 m |
| Telescope planar FOV (full width) | 165 μrad |
| Telescope reflectivity | 0.89 |
| Fiber transmission | 0.82 |
| Filter transmission | 0.76 |
| Receiver transmission (includes 10% losses for second channel) | 0.84 |
| Detector quantum efficiency | 0.50 |

**Table 2.** Instrument parameters used in the performance simulations.

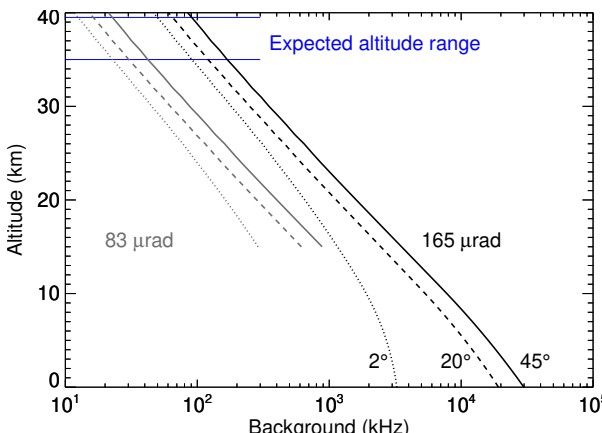

**Figure 5.** Simulated background photon counting rates for the telescope FOVs 83 μrad (gray) and 165 μrad (black) and solar elevation angles 88°, 70° and 50°. The 165 μrad FOV was eventually used during the mission.

added to the simulated lidar profile. A detailed description of the simulation and underlying equations is presented in the appendix.

Figure 5 shows background photon counting rates for three different solar elevation angles as function of altitude. 45° represents the maximum solar elevation which occurs at solar noon on 24 December at 68° southern latitude. Because long-duration balloons launched from McMurdo remain generally southward of 68°S and solar elevation angles at solar noon are lower before and after 24 December, the 45° trace in Figure 5 can be seen as worst case estimate for the Rayleigh background. However, it is important to note that the simulation includes a telescope with an unobstructed view of the sky, whereas in reality

the spider of the telescope is within the telescope FOV and photons scattered off the spider vanes enhance the background counting rates. How much the spider contributes to the total background depends on several factors (e.g. exact geometry,





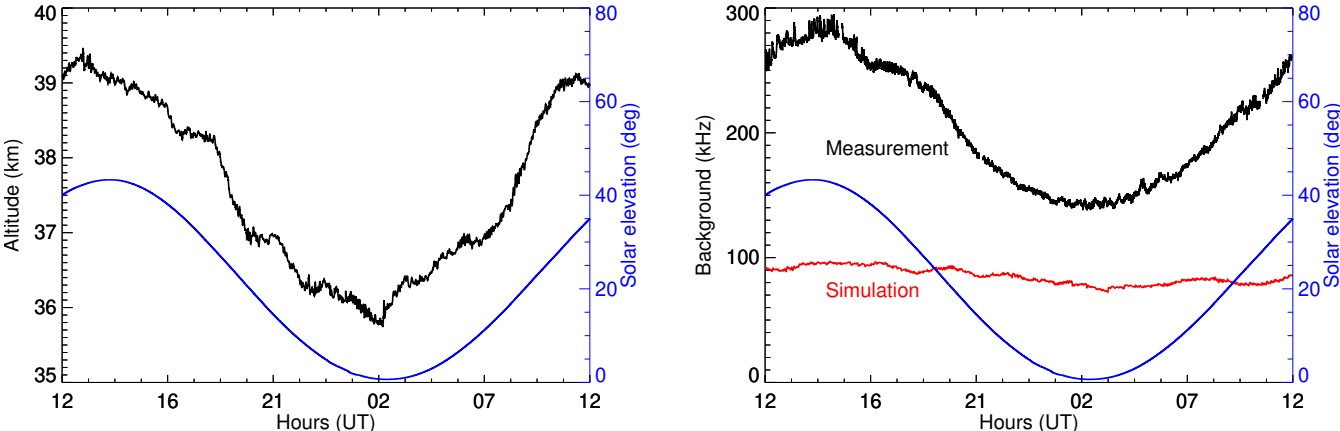

**Figure 6.** GPS altitude of the gondola (left) and measured and simulated lidar background (right) between the evening of the 10th and morning of the 11th July 2018. The simulation (red line) includes the actual altitudes and solar elevation angles.

surface reflectivity, effective sky brightness) which are difficult to estimate. Therefore, we defer the discussion to the discussion section (Section 5).

The 2° trace in Figure 5 marks the Rayleigh background for solar midnight at 70°S on 1 December. This is about the

earliest launch opportunity for a long duration balloon, and later launches result in larger minimum solar elevation angles and hence a larger background. Thus, the area between the 2° and 45° lines characterizes the maximum potential variation of the Rayleigh background during the originally planned Antarctic mission. The flight track of the eventually executed mission remained within the same lattitude band albeit in the northern hemisphere, and thus the same estimates apply. Considering floating altitudes between 35 and 39.5 km, the expected Rayleigh background lies between 50 kHz and 180 kHz. It is worth

mentioning here that the floating altitude of zero-pressure balloons varies by 3-4 km in response to solar heating, i.e. the balloon rises in the morning and descends in the evening (see Figure 6). On the one hand greater heights result in lower background rates as shown in Figure 5, and on the other hand larger solar zenith angles cause larger sky brightness and hence larger background rates. Because the variations in altitude and in solar zenith angle are in phase, the two effects roughly cancel each other and the simulated Rayleigh background (red line in Figure 6) is rather constant at approximately 90 kHz. However, actual

measurements of the total background rate are 2-3 times larger and follow the trend of the solar elevation.

Figure 7 shows measured and simulated photon count profiles for a 30 s integration period on 11 July 04:07 UTC. In order for the simulation to reproduce the measured profile, the transmission of the receiver was artificially reduced by 38 % and the background increased by 280%. This assumes that an additional loss of signal occurs somewhere in the receiver and thus affects both the lidar return signal and the background in the same way. In order to assess the detection threshold of the

BOLIDE instrument, we calculated the backscatter ratio and the volume backscatter coefficient of the PMC ($\beta_{PMC}$) using the NRLMSISE-00 density profile of the atmosphere for July and 69°N latitude as reference density profile (Picone et al., 2002). The PMC signal in Figure 7a originates from a rather bright cloud with a peak $\beta_{PMC}$ of $42 \times 10^{-10} \mathrm{m}^{-1}\mathrm{sr}^{-1}$. We used this



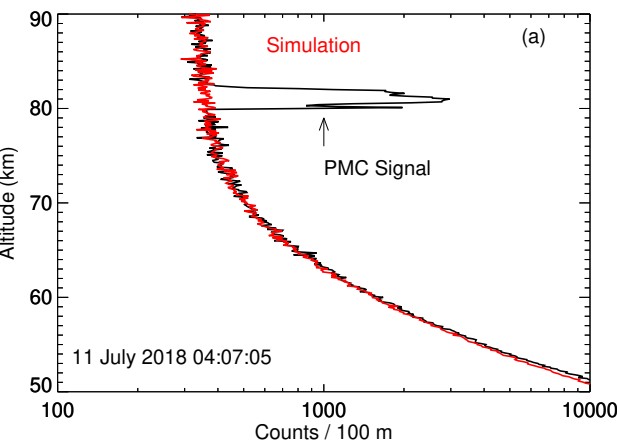
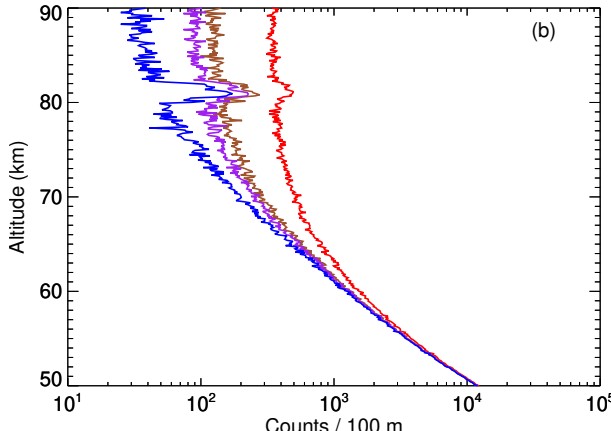

**Figure 7.** Measured (black) and simulated (colored) photon count profiles for a 30 second integration period on 11 July 2018 04:07 UTC. The peak $\beta_{\mathrm{PMC}}$ in (a) is $42 \times 10^{-10}\mathrm{m}^{-1}\mathrm{sr}^{-1}$. (b) shows simulated profiles with a peak $\beta_{\mathrm{PMC}}$ of $2 \times 10^{-10}\mathrm{m}^{-1}\mathrm{sr}^{-1}$ for the configuration 165 µrad FOV and high background (red), 165 µrad FOV with Rayleigh background only (brown), 83 µrad FOV with high background (purple), and 83 µrad FOV with Rayleigh background only (blue).

profile scaled to a peak value of $2 \times 10^{-10}\mathrm{m}^{-1}\mathrm{sr}^{-1}$ as reference profile for the following simulations shown in Figure 7b: The red profile represents the configuration as flown with a telescope FOV of 165 µrad and a background which is 2.8 times the

Rayleigh background. For the brown profile we assume that the additional background can be avoided through better shielding of the telescope i.e. we consider only the Rayleigh background. This is a pathological case in the sense that the spider which is located within the FOV of the telescope will always contribute some additional background. Finally, the background can be lowered further if we reduce the telescope FOV to 83 µrad by using a smaller optical fiber. The impact of this smaller FOV is demonstrated by the last two curves in Figure 7b: For the purple profile we assume again a background which is 2.8 times

the Rayleigh background and thus represents a conservative estimate assuming that the shielding of the telescope cannot be significantly improved. In contrast, if the telescope can be perfectly shielded, we end up with the blue profile. Latter represents the optimum performance and technological limit of the BOLIDE instrument. Following Kaifler et al. (2013) we define a PMC as detectable if the PMC signal with the background removed is larger than 2.5 times the measurement uncertainty (1 sigma error). We find detection thresholds of 0.58, 0.38, 0.33 and 0.19 in units of $10^{-10}\mathrm{m}^{-1}\mathrm{sr}^{-1}$ for the four simulated profiles

shown in Figure 7b. The former value (0.58) applies to the flight configuration and thus represents the detection threshold in the acquired data. In total, the instrument recorded backscatter profiles of PMCs ($\beta > 2 \times 10^{-10}\mathrm{m}^{-1}\mathrm{sr}^{-1}$) during 70 h out of the $\sim 5.9$-day flight.

Temperature profiles were retrieved by hydrostaic integration of photon count profiles below the PMC layer (Hauchecorne and Chanin, 1980). Here, the excessive background noise becomes strongly noticeable and limits effectively the resolution of

retrieved temperatures to 1.3 km in the vertical and 20 min in time. Examples of BOLIDE temperature data are shown in Fritts et al. (2019). The retrieval of temperature profiles above the PMC layer as demonstrated in Kaifler et al. (2018) is not possible due to the high background and the resulting poor signal-to-noise ratio above the layer.





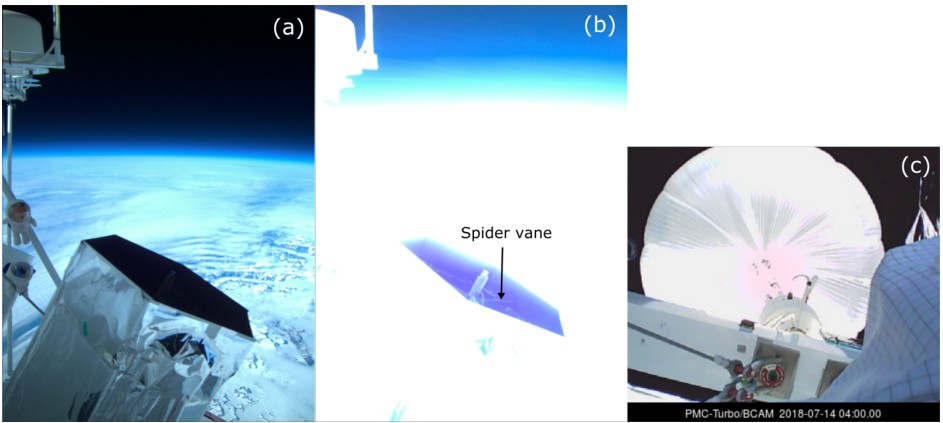

**Figure 8.** Normal (a) and overexposed (b) image of the upper part of the telescope taken by the side viewing onboard camera. Image (c), taken by an upward looking engineering camera inside the pressure vessel, shows the balloon above the gondola.

## 5    Discussion

As demonstrated in Figure 7 and in more detail in the overview paper by Fritts et al. (2019), lidar measurements acquired by
the BOLIDE instrument during the PMC Turbo mission yielded high resolution backscatter profiles of PMC. All requirements
(see Table 1) were met and the performance surpassed that of the largest ground-based lidar systems. When comparing the
actual performance to pre-flight predictions, we found two major discrepancies: 1) The detected lidar signal was 38 % lower
and 2) the background was a factor of 2-4 higher. The latter number is a rough estimate because the answer to the question of
how much additional background BOLIDE detected depends ultimately on where the unexpected signal loss occurred. Based
on post-flight tests, the most likely cause of the signal loss is a misalignment between the last lens of the optical system and
the APD detector (see Figure 2). This particular lens has a focal length of 8 mm and, for optimal detection efficiency, has
to be precisely aligned to the optical axis of the detector. Because the active area of the APD is only 180 μm in diameter,
any even small misalignment leads to significant losses which affect the lidar signal and the background in exactly the same
way. If, however, the detected signal loss was caused by incomplete overlap between the telescope FOV and the laser beam,
for example as a result of a much larger beam divergence, then only the lidar signal would have been affected. Thus, in this
scenario the relative increase in background relative to pre-flight predictions would be lower as compared to the previous case
where lidar signal and background are affected in the same way. Due to this uncertainty, we cannot put narrower constraints on
the background.

The unexpectedly high background was identified as a potential problem in the early design phase of BOLIDE. As evident
from Figure 8C, the balloon above the gondola acts as a bright and diffuse light source which illuminates the gondola and the
telescope. Because of the height requirement (Table 1), there was no way to extend the baffle of the telescope and shield the
spider vanes from light scattered off the balloon. Indirect illumination where the light scatters off the inside of the telescope
tube first and then hits the underside of the spider vane is also possible, but was not investigated during the design phase due




to time and financial constraints as well as due to lack of a clear understanding of the optical properties of the balloon. Images taken during the flight documented that the spider vanes are indeed significantly illuminated by the balloon. While the inside of the telescope appears pitch black against bright sunlit tropospheric clouds (Figure 8a), the overexposed image reveals the spider vanes and the focusing assembly inside the telescope (Figure 8b). That is only possible if the inside of the telescope is illuminated from above. Rayleigh scattering in the atmosphere above the gondola can be ruled out as major source because the radiance as seen from the gondola is roughly constant over a period of 24 hours, whereas the measured background shows a

clear dependence on the solar elevation angle (see Figure 6). Hence, the balloon must be the dominating light source.

There are two obvious solutions to the background problem: the addition of a baffle to prevent stray light from entering the telescope and the use of an off-axis telescope. As noted above, the former was not possible for the PMC Turbo mission because of height constraints, but can be accommodated in a future mission with a redesigned gondola. The use of an off-axis telescope is an elegant but costly solution. Because such a telescope does not require a spider, there is no obstruction within the

FOV of the telescope and thus no possibility for stray light entering the optical path except for scattering at the mirror surface. Reducing the telescope FOV from 165 µrad to 83 µrad helps in both ways as it not only decreases the additional background due to the smaller acceptance angle, but also reduces the Rayleigh background (see Figure 7). A FOV of 83 µrad is technically possible with the BOLIDE system, though in this case the laser beam fills nearly completely the telescope FOV with little margin. Since we had no previous experience with operating a lidar on a balloon, we were not sure about the mechanical and

thermal stability of the laser-telescope assembly and the vibration environment at floating altitude. Therefore, we opted for a conservative approach and used the larger FOV which tolerates larger excursions of up to ~40 µrad off the nominal beam position without degradation of the lidar return signal. Tests carried out during the flight suggested a low vibration environment and very stable beam pointing. Hence, in future flights of the BOLIDE instrument we plan to use the small FOV. As evident from the simulations shown in Figure 7, the small FOV in combination with better shielding or an off-axis telescope will push

the background to a value below the Rayleigh scattering at the PMC altitude. In that case the PMC detection threshold will be determined by the Rayleigh scattering rather than the background and no further improvements are possible unless the laser power or the telescope aperture or both are increased.

From an operational point of view the instrument performed well throughout the flight. No degradation of the optics was observed and signal levels remained constant. The flight software worked as expected with one exception: A bug in the data

acquisition software of the lidar caused the data acquisition to stop after approximately 12 hours, and the only way to exit this condition was a manual restart. Although the capability to patch the flight software in flight was built into the software, we did not make use of it because the stop times were predictable and restarts could be initiated via real-time or time-tagged commands. Later, the inspection of the source code revealed that the bug would have been caught before the flight with longer integrated testing of the hard and software, as pre-flight testing had been limited to continuous runs of about 8 hours. The

downlink of real-time science data proved to be valuable for the assessment of science conditions and PMC brightness and, in particular during the early hours of the flight, served as guidance for the imaging experiments. We also made frequent use of the commanding capability to adjust settings and thus optimize the performance of the instrument.





## 6 Conclusions

As noted above, the BOLIDE experiment turned out to be highly successful during its maiden six day long flight in July 2018.
It met all design requirements and provided science data of PMCs with better quality (higher resolution and lower detection
threshold) than ground-based lidar experiments. The comparison of in-flight performance and predicted performance revealed
that the instrument suffered from excessive background noise caused by light scattered off the balloon entering the optical path.
Two obvious solutions for this problem are the addition of a baffle to prevent stray light from entering the telescope and the
use of an off-axis telescope.

BOLIDE also demonstrated that it is possible for a small team of dedicated scientists and engineers to design, develop and
build the flight hardware within a short period of about two years. In addition to providing science data, the primary goal of
the BOLIDE project was to demonstrate for the first time the operation of a high-power lidar on board a balloon platform.
In this regard the authors hope that the successful operation of BOLIDE paves the way for future more capable and more
ambitious balloon lidar experiments. An obvious candidate is a compact iron resonance lidar for measuring high altitude winds
and temperatures (Kaifler et al., 2017).

*Data availability.* Data are available on the HALO database https://halo-db.pa.op.dlr.de/mission/112

## Appendix A: Equations used in the Simulation

The Rayleigh signal per laser pulse at altitude $z$ integrated over the height of an altitude bin $\Delta z$ is given by

$$S_{\mathrm{Ray}}(z) = \Delta z \rho(z) \sigma_{180}(\lambda) \frac{A}{(z - \bar{z})^2} N, \tag{A1}$$

where $\rho(z)$ is the number density taken from the NRLMSISE-00 model (Picone et al., 2002), $\sigma_{180}(\lambda)$ the Rayleigh backscatter
cross-section, and $A(z - \bar{z})^{-2}$ the solid angle formed by the aperture of the telescope with area $A$ and distance $z - \bar{z}$. Note that
$\bar{z}$ describes the altitude of the telescope i.e the floating altitude of the gondola. The number of photons emitted per laser pulse
is expressed as

$$N = \frac{E\lambda}{hc} \tag{A2}$$

with the pulse energy $E$, Plank's constant $h$, and the speed of light $c$. For the wavelength $\lambda$ of our laser we used the value
532.32 nm (vacuum wavelength). The backscatter cross-section is computed via:

$$\sigma_{180}(\lambda) = \frac{1}{4\pi} \frac{3}{2} \sigma(\lambda) \tag{A3}$$

The factor $3/2$ originates from the Rayleigh phase function for the scattering angle of $180°$, and the total scattering cross-
section $\sigma = 5.16 \times 10^{-31} \mathrm{m}^{-2}$ is calculated from the equations given in Bucholtz (1995).





We simulated the solar Rayleigh background based on sky radiances calculated for given altitudes $\bar{z}$ and the looking direction anti-sun $28°$ off zenith using the libRadtran software packet for radiative transfer calculations (Emde et al., 2016). The radiances $L_{\mathrm{e},\Omega}$ were converted to received power as:

$$P(\bar{z}) = L_{\mathrm{e},\Omega}(\bar{z}) A\Omega B \tag{A4}$$

where $\Omega = \pi(0.5 \cdot \mathrm{FOV})^2$ is the solid angle of the telescope with the field of view FOV. $B$ is the transmission bandwidth of the
interference filter in the receiver (see Figure 2). In general one would have to integrate the product of wavelength-dependent radiance and filter transmission over the full spectrum to obtain the received power. Because there are no Frauenhofer lines within the passband of our 0.3 nm wide filter, for simplicity we assumed a constant radiance and replaced the transmission function with a corresponding rectangular function with the same area as the transmission function integrated over the full spectrum. In our case we obtained $B = 0.250$ nm. In a next step, the Rayleigh background photon rate can be expressed in
analogy to Eq. A2 as

$$n_{\mathrm{BG}} = \frac{P(z)\lambda}{hc}. \tag{A5}$$

After multiplying by $\tau = 2\Delta z c^{-1}$, which is the period of a range gate, the resulting background photons per range gate can be added to the Rayleigh signal to form the total photon profile. Multiplication with the number of laser pulses per integration period, $\Delta t f_{\mathrm{rep}}$, yields the the received photon count profile

$$S(z) = \eta \Delta t f_{\mathrm{rep}} \left( S_{\mathrm{Ray}}(z) + \tau n_{\mathrm{BG}} \right). \tag{A6}$$

Here $\Delta t$ is the integration period and $f_{\mathrm{rep}}$ the pulse repetition frequency of the laser. The prefactor $\eta$ in equation A6 is the efficiency of the lidar system and is computed as product of all efficiencies and transmission coefficients listed in Table 2. Taking into account the additional 38 % signal loss in the receiver (see Sect. 4), we used $\eta = 0.191$ as final value.

The photon count profile described by equation A6 is valid for a perfect noise-free system. In order to simulate the shot-noise
of the photon counting process, in a final step we replaced each count value of $S(r)$, i.e. the number of detected photons for a given range gate, with a random number drawn from a Poisson distribution where the expected value of the distribution equals the original count value.

*Author contributions.* BK came up with the idea for the BOLIDE instrument, built the receiver, managed the project and wrote this manuscript. DR designed the pressure vessel and telescope. CB built the laser transmitter system. NK wrote the software for the instru-
ment and performed data analysis. VB designed the thermal control system.

*Competing interests.* The authors declare that they have no conflict of interests.



*Acknowledgements.* Development of the BOLIDE instrument was supported by DLR. The gondola was funded partly by NASA and DLR and was built by the PMC Turbo Team with support from NASA. The Authors thank David C. Fritts and NASA for the flight opportunity, Markus Rapp for acquiring funding, and the staff of the Columbia Scientific Balloon Facility for their excellent support during preparation

and execution of the mission.



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
