# Peer review of "A technical description of the Balloon Lidar Experiment BOLIDE"

_Atmospheric Measurement Techniques, 2020_

## Referee Comment (RC1) · Anonymous Referee #1 · 1 Jun 2020

Review of "A technical description of the Balloon Lidar Experiment BOLIDE" by Kaifler et al. The paper give an excellent overview of the design and development of a novel Rayleigh lidar system operated on broad a NASA high altitude balloon mission. This includes the choice of laser system and telescope, especially the team's work on reducing the optical background noise. The match of FOV between the telescope and the laser beam diameter seems to be the critical part of reducing the background. The lidar data prove the system achieved its major capabilities and worked during the flight as designed, although there is room to improve the system in the future mission. The documented experience in this paper is extremely valuable for future balloon or spaceborne lidar design work. I would appreciate the author to clarify on the FOV technique, which I do not fully understand. The fiber is 200 ïA■m, 0.22 NA and the

focal length of the telescope is f=2.4*0.5=1.2 m. If we reverse the ray tracing, the only return signals that can be reflected into the fiber are those centered within 265.6 mm diameter disk of the telescope. If this is the case, there would be no need to use 0.5 diameter telescope. From another perspective, the conservation of etendue, if we assume the divergence of the laser beam is 70 ïA■rad, the etendue of the object illuminated atmosphere at 70 km by the laser and received by the 0.5 m telescope is about (0.5/70,000)^2*(5 m)^2. The etendue of the fiber is (0.22 rad)^2*(200ïĆť10-6 m)^2, which is much less. Am I missing something here? Could the author also please explain why three detectors (two APDs and one PMT) are set up in the receiver? Minor comments: Line 20, please consider to replace "wanted" with "desired" Line 23, please consider to replace the reference Wickwar et al., 2016 with Sox et al., 2018 (a JRG paper for the same topic). Line 27, please consider to add "Krueger et al., 2015" as the reference on Na Doppler lidar. Line 35, One of the other advantage of the balloon lidar is that The high altitude flying lidar also avoids the scarified laser power and return signal due to strong scattering in the troposphere, suffered by the ground-based lidar system. Line 129, please consider to replace "Not converted" with "The residual". Line 258, please consider to change the phrase "worst case estimated" to "worst case scenario estimation". Line 262-263, the last sentence of the paragraph "Therefore, we defer the discussion to the discussion" sounds a little strange, please consider to rephrase. Line 277-278, please explain the underline motivation to choose these two factors, 38% and 280% Line 324, please consider to delete "due to"

Please also note the supplement to this comment:
https://www.atmos-meas-tech-discuss.net/amt-2020-150/amt-2020-150-RC1-supplement.pdf

---

## Author Comment (AC1) · 2 Jun 2020

We thank the reviewer for pointing out shortcomings in the description of the lidar system. We will address those in the following as well as in the revised manuscript. We also thank the reviewer for making suggestions for better wording, which we will incorporate in the revision.

**1   Field of view of the telescope and overlap with the laser beam**

Matching of the field of view of the telescope (FOV) to the divergence of the laser beam is indeed critical for achieving low background noise and thus a high signal to

noise ratio. In the sunlit atmosphere, scattered sunlight dominates the background noise in lidar data. As the strength of the solar background is proportional to the FOV of the telescope, one aims to make the FOV as small as possible. The constraint here is that the FOV must be at least as large as the divergence of the laser beam. Making it any smaller than that still reduces the background but at the same time also reduces the amount of scattered laser light collected by the telescope. With a FOV smaller than the divergence of the laser beam, the telescope can "see" only a fraction of the beam and the net result is a decrease in the signal to noise ratio. Thus, the maximum signal to noise ratio that can be achieved depends ultimately on how small you can make the laser beam divergence. The use of a beam expander trades the reduction of the divergence against an increase in the beam diameter at the output of the expander. In our case, the limiting factor was the clear aperture ($\sim 50$ mm diameter) of the window in the pressure vessel through which the beam had to pass before being transmitted into the sky. Given the maximum possible beam diameter in our setup, the resulting full angle beam divergence was $77$ µrad. This in turn set a lower limit to the FOV of the telescope to approximately $87$ µrad. We included a margin of $5$ µrad on either side to account for potential misalignment between the optical axes of the laser beam and the telescope as a result of vibrations or other drifts.

The FOV of the telescope is determined by the focal length $f$ of the telescope and the diameter of the fiber core $d$ mounted in its focal plain:

$$\text{FOV} = \frac{d}{f} \tag{1}$$

With $f = 2.4 \cdot 505$ mm and $d = 0.2$ mm the above equation yields FOV $= 165$ µrad. Choosing a $100$ µm diameter fiber results in a FOV of $82.5$ µrad. We deemed that as too small as this would leave less than $3$ µrad on either side for potential misalignment between the optical axis of the telescope and the laser beam.

As pointed out by the reviewer, the numerical aperture (NA) of the fiber needs to match

the NA of the telescope in order to achieve an efficient coupling. The NA is defined as

$$NA = n \sin \Theta_{\text{max}} \tag{2}$$

where $\Theta$ is the *half angle*, i.e. the angle between the optical axis and the ray, and $n$ is the index of refraction. Using above equation, the NA of the telescope is given by $\sin(0.5 \, D/f) = 0.197$ ($D = 505$ mm is the diameter of the telescope mirror). The NA of the telescope is thus slightly smaller than the NA of the fiber (0.22) as illustrated in Figure 1, meaning the fiber accepts all rays reflected off the mirror and not just those coming from within a circle of $256$ mm diameter as claimed by the referee. Maybe the referee used the full angle in equation 2 instead of the half angle.

**2  Rationale for using three detectors**

A general problem with mesospheric Rayleigh lidar systems is the huge dynamic range of the lidar return signal. One factor is the exponential decrease in air density (the density changes by approximately 6 orders of magnitude from ground to 100 km altitude), and the other factor is the $r^2$ dependence of the scattered light. The result is a very strong lidar signal coming from the lower atmosphere with peak counting rates of Hundreds of MHz. Most single photon detectors are good up to counting rates of 5-10 MHz. For higher counting rates the signal becomes increasingly nonlinear and thus unusable.

A pragmatic solution to this problem is the use of multiple gated detectors where each detector sees only a fraction of the signal. A commonly used splitting ratio is 1:10. In this case the second detector sees a tenfold lower signal as compared to the first detector, and the signal provided by the second detector may be still usable in an altitude range where the signal from the first detector is already in the strongly nonlinear regime. This works until the lidar signal becomes so strong that the output of the second detector becomes strongly nonlinear and a third detector may have to be used.
In our case a single detector was sufficient to handle the PMC backscatter signal. However, with the BOLIDE instrument we also wanted to measure Rayleigh temperature profiles below the PMC layer, reaching down as close as possible to the balloon gondola. Figure 2 shows photon count profiles from all three detectors (APD1 (black), APD2 (red), PMT (blue); the gondola was at approximately 37 km altitude). It turned out that defocusing of the telescope in the near field rendered the lower 5-8 km of the profiles useless and, as a result, the third detector exhibited much lower counting rates at low altitudes than anticipated.

**3 Motivation for choosing the factors 38 % and 280 %**

When all known factors (e.g. laser power, telescope aperture, transmittance of filters) were taken into account, our simulations produced a simulated lidar signal which was larger than what our instrument detected in flight. However, when we decreased the simulated signal by 38 %, simulation and actual measurements were in good agreement. This let us conclude that we have unknown losses somewhere in the optical path, most likely due to a misalignment of the last lens in front of the detector. These losses amount to 38 % of the total signal, as estimated from the comparison between simulated and measured signal.

Likewise we simulated the solar background which is supposedly detected by our instrument and compared those simulations with our measurements. It turned out that we had to increase the simulated background by 280 % in order to bring it into agreement with the measurements. Our interpretation is that additional scattered sun light was collected by the BOLIDE instrument via a process not included in our simulations. A likely candidate is scattering of diffuse sun light of the telescope spider.

[Figure]

Telescope f/2.40
D = 0.50 m
f = 1.212 m
NA = 0.197

Fiber
D = 0.20 mm
NA = 0.220

FOV
0.165 mrad

Telescope angle

Fiber angle

Mirror

Distance (m)

Distance (m)

**Fig. 1.**

**Fig. 2.**

---

## Referee Comment (RC2) · Anonymous Referee #2 · 9 Jun 2020

The manuscript decribes the design, development and first results of a novel lidar operated on a NASA long duration balloon mission. The lidar operated for 6 days in July 2018 and sucessfully recorded Polar Mesospheric Clouds (PMC). The manuscript describes the technical aspects of the lidar development and performance estimations for the detection of PMC. The estimates are compared to one example on July 11, 2018. The paper is well written and allows for an easy understanding of the complicated topic. It was a pleasure to read the manuscript.

I only have some minor comments in addition to those by Referee #1:

Page 4, Line 101: "180 $\mu$rad" → "165 $\mu$rad"?

Page 6, Line 136: "First mirror" → "Motorized mirror"

[Figure]

Page 6, Line 146: "100 Hz rotations..." what is the actual chopping frequency? For balance reasons I would guess the chopper blade has more than 1 hole.

Page 9, Line 197: what are "the laser replacement heater and radiator bias heaters"? Does that statement mean the same thing as the "replacement heaters" in line 191?

Page 11, Lines 255 to 265: For me the detailed discussion of the plans for launches from Mc Murdo seem disturbing. At the end the summary is that $2°$ to $45°$ is the range of relevant solar elevation angles. Fig. 6 then shows solar elevation angles between $<1°$ and $\sim40°$, I see no discrepancy here, but the text was just slightly confusing to read.

Page 12, Line 282: "We used this profile scaled ..." → "We used the profile of $\beta_{PMC}$ scaled ..."

Page 306 and 362: "... surpassed that of the largest ground-based lidar systems" and "... with better quality (higher resolution and lower detection threshold) than ground-based lidar experiments."

It is unclear what the actual parameters of the ground-based lidar systems are. I guess the statement is probably true at times, but I also guess that the performance of a ground based lidar varies substantially throughout the day, even more than the variation seen in BOLIDE (Fig. 6).

I suggest revisiting this statement, probably with updating the reference to "ground-based lidar systems" given on Page 3, line 79: There is one reference given to a rather old publication (Fiedler et al., 2011) where a decadal scale dataset was analized with a resolution that was for this purpose chosen to be 14 minutes. Since then a couple of publications have shown lidar observation of NLC with a resolution higher than 14 minutes and some with resolutions well below 30 seconds and 100 m. Measurement uncertainties have been reported as low as $0.1 \times 10{-}10$ m$-1$ sr$-1$.

Kaifler et al., JGR, 2018; https://doi.org/10.1029/2018JD029717

Ridder et al., JASTP, 2017; http://dx.doi.org/10.1016/j.jastp.2017.04.005

Fritts et al., JASTP, 2017; http://dx.doi.org/10.1016/j.jastp.2016.11.006

Kaifler et al., ACP, 2013; http://dx.doi.org/10.5194/acp-13-11757-2013

Baumgarten, GRL, 2012; http://dx.doi.org/10.1029/2011GL049935

---

## Referee Comment (RC3) · Anonymous Referee #3 · 29 Jun 2020

This paper describes the BOLIDE balloon-borne lidar experiment for the detection of polar mesospheric clouds. The advantage of being on a balloon is a signal gain due to the shorter distance between the lidar and the clouds, the guarantee of always having clear sky conditions and the reduction of background light during the day. This lidar is built to be installed on a balloon pod with other instruments to fly in the Arctic or Antarctic. It flew for 6 days in the Arctic region during the PMC turbo campaign in July 2018. A troubling point is that the document only describes the project to fly the lidar from Mac Murdo in Antarctica but it flew from Kiruna in the Arctic. This change of flight location needs to be explained. The document is more of a technical report describing the instrument and its performance than a scientific document. For example, the description of the lidar subsystems in section 3 is very long and rather technical. I

suggest that we limit ourselves to the essential elements of the document and move the detailed description to the appendix. There are almost no scientific results presented in the document. Some results from BOLIDE lidar during the PMC-Turbo campaign were presented in Fritts et al (2019) but it would be useful for readers to show some examples of observations of polar mesospheric clouds and Rayleigh temperature.

————————————————————

---

## Author Comment (AC2) · 8 Aug 2020

We thank the reviewer for the detailed and careful analysis of our paper and for suggestions for improved wording. We will address all comments in the revision of the manuscript. Answers to specific questions and comments are given below.

**What is the actual chopping frequency of the chopper?**

The chopper blade (104 mm diameter) has three slits 60 degrees in length. At 100 Hz rotation rate, the chopping frequency is thus 300 Hz.

**What are laser replacement heater and radiator bias heaters?**

From a thermal management point of view it is desirable to keep the heat load on the cooling system constant. On the one hand this simplifies the control system and avoids having to deal with transients. On the other hand, the thermal analysis, which was done when designing the instrument, is based on steady state solutions. Large transients may drive the cooling system beyond its design parameters. That should be avoided.

However, there are certain phases during the mission when the instrument needs to be reconfigured, e.g. the laser needs to be switched off. In order to keep the head load constant, whenever a component is switched off so-called replacement heaters are activated. These heaters are designed such that they dissipate as much electrical power as the components they replace. By activating replacement heaters, the total power dissipation and thus the heat load is held constant.

Still there are slow variations in the performance of the radiator. Most notably, in the course of the day the solar elevation angle changes, and this results in a varying amount of scattered solar radiation hitting the radiator surface. With increasing absorbed solar power, the total power (nominal heat generated by the instrument + absorbed heat from the environment) the radiator needs to reject also increases. Since the radiated power is proportional to the temperature to the fourth power, the coolant temperature rises when the total power increases. To counter this effect and keep the coolant temperature constant, the radiator was designed with some margin in its heat rejecting ability and electrical heaters mounted at the backside of the radiator were incorporated into the design. At low solar elevation angles, the heaters are powered to increase the temperature of the radiator surface. Hence the name bias heaters - the heaters bias the temperature of the radiator surface.

**It is unclear what the actual parameters of the ground-based lidar systems are**

That is true, we did not define what we mean by this statement. The instrument requirements were defined in 2015 when the PMC Turbo mission was designed. At that time it was decided that the BOLIDE instrument should have at least the same sensitivity as the largest ground-based instruments, so that the same science in terms of resolution and detection threshold can be accomplished with BOLIDE as what was then possible with ground-based instruments. The study by Fiedler et al. (2011) served as a reference for this purpose.

We acknowledge that there are more recent publications demonstrating lower detection thresholds and/or higher resolutions. As noted by the reviewer, e.g. Kaifler et al. (2018) report the instrument sensitivity $0.1 \times 10^{-10} \mathrm{m}^{-1} \mathrm{sr}^{-1}$. However, these papers were published after the BOLIDE requirements were defined. Moreover, in case of Kaifler et al. the very high performance was achieved only during a few hours in darkness, while BOLIDE acquired measurement of PMC with approximately constant performance continuously for several days.

We will change the text to make it clear that we compare the performance of BOLIDE to ground-based observations which were published before the design phase of BOLIDE.

**References**

Fiedler, J., Baumgarten, G., Berger, U., Hoffmann, P., Kaifler, N., and Lübken, F.-J.: NLC and the background atmosphere above ALOMAR, Atmospheric Chemistry and Physics, 11, 5701–5717, https://doi.org/10.5194/acp-11-5701-2011, https://www.atmos-chem-phys.net/11/5701/2011/, 2011.

Kaifler, N., Kaifler, B., Wilms, H., Rapp, M., Stober, G., and Jacobi, C.: Mesospheric Temperature During the Extreme Midlatitude Noctilucent Cloud Event on 18/19 July 2016, Journal of Geophysical Research: Atmospheres, 123, 13,775–

13,789, https://doi.org/10.1029/2018JD029717, https://agupubs.onlinelibrary.wiley.com/doi/abs/10.1029/2018JD029717, 2018.

---

## Author Comment (AC3) · 8 Aug 2020

We thank the reviewer for her/his comments which we address below.

**Change of flight location**

As pointed out by the reviewer, there is information missing in the manuscript. When the PMC Turbo mission was conceived, the payload was designed for circumpolar flight in Antarctica. Less than a year before the anticipated launch date it became clear that a launch from McMurdo, Antarctica would not be possible within the funding period due to a backlog of payloads waiting for a launch opportunity in Antarctica. Instead, NASA

offered a 6-day flight in the northern hemisphere with launch from Esrange, Sweden. An analysis showed that the BOLIDE instrument was compatible with such a flight in the northern hemisphere. But since the instrument was really designed for Antarctica, e.g. the filter wheel would only be needed in McMurdo and not at Esrange, in this manuscript we decided to report on the original mission and mission requirements as much as possible.

We will add this background information in the revised manuscript.

**The document is more of a technical report describing the instrument and its performance than a scientific document**

That was exactly our intention and is hopefully conveyed by the title of the manuscript. First scientific results are published in Fritts et al. (2019), Fritts et al. (2020) and Geach et al. (2020). More scientific papers are submitted and will be published soon.

Because the technical description of the instrument and performance analysis is the heart of the paper, we prefer to keep the current structure and not move the description to the appendix as suggested by the reviewer. However, we will add following figure to Figure 7 showing a height-time section of the PMC backscatter signal.

**References**

Fritts, D. C., Miller, A. D., Kjellstrand, C. B., Geach, C., Williams, B. P., Kaifler, B., Kaifler, N., Jones, G., Rapp, M., Limon, M., Reimuller, J., Wang, L., Hanany, S., Gisinger, S., Zhao, Y., Stober, G., and Randall, C. E.: PMC Turbo: Studying Gravity Wave and Instability Dynamics in the Summer Mesosphere Using Polar Mesospheric Cloud Imaging and Profiling From a Stratospheric Balloon, Journal of Geophysical Research: Atmospheres, 124, 6423–

[Figure]

**Fig. 1.** (c) PMC backscatter $\beta_{\mathrm{PMC}}$ in units of $1 \times 10^{-10}\mathrm{m}^{-1}\mathrm{sr}^{-1}$ shown for a 4-hour long period centered around the profile in (a).

6443, https://doi.org/10.1029/2019JD030298, https://agupubs.onlinelibrary.wiley.com/doi/abs/10.1029/2019JD030298, 2019.

Fritts, D. C., Kaifler, N., Kaifler, B., Geach, C., Kjellstrand, C. B., Williams, B. P., Eckermann, S. D., Miller, A. D., Rapp, M., Jones, G., Limon, M., Reimuller, J., and Wang, L.: Mesospheric Bore Evolution and Instability Dynamics Observed in PMC Turbo Imaging and Rayleigh Lidar Profiling Over Northeastern Canada on 13 July 2018, Journal of Geophysical Research: Atmospheres, 125, e2019JD032 037, https://doi.org/10.1029/2019JD032037, https://agupubs.onlinelibrary.wiley.com/doi/abs/10.1029/2019JD032037, e2019JD032037 2019JD032037, 2020.

Geach, C., Hanany, S., Fritts, D., Kaifler, B., Kaifler, N., Kjellstrand, C., Williams, B., Eckermann, S., Miller, A., Jones, G., and et al.: Gravity Wave Breaking and Vortex Ring Formation Observed by PMC Turbo, Earth and Space Science Open Archive, p. 23, https://doi.org/10.1002/essoar.10503122.1, https://doi.org/10.1002/essoar.10503122.1, 2020.

---

## Author Response (AR1)

We thank the reviewers for their comments and suggestions which helped us to improve the manuscript. The comments of three anonymous reviewers are reproduced in blue and author responses are written in black.

**Reviewer #1**

Review of "A technical description of the Balloon Lidar Experiment BOLIDE" by Kaifler et al.

The paper give an excellent overview of the design and development of a novel Rayleigh lidar system operated on broad a NASA high altitude balloon mission. This includes the choice of laser system and telescope, especially the team's work on reducing the optical background noise. The match of FOV between the telescope and the laser beam diameter seems to be the critical part of reducing the background. The lidar data prove the system achieved its major capabilities and worked during the flight as designed, although there is room to improve the system in the future mission. The documented experience in this paper is extremely valuable for future balloon or spaceborne lidar design work.

We appreciate the assessment.

I would appreciate the author to clarify on the FOV technique, which I do not fully understand. The fiber is 200 μm, 0.22 NA and the focal length of the telescope is f=2.4*0.5=1.2 m. If we reverse the ray tracing, the only return signals that can be reflected into the fiber are those centered within 265.6 mm diameter disk of the telescope. If this is the case, there would be no need to use 0.5 diameter telescope. From another perspective, the conservation of etendue, if we assume the divergence of the laser beam is 70 μrad, the etendue of the object illuminated atmosphere at 70 km by the laser and received by the 0.5 m telescope is about $(0.5/70,000)^2*(5m)^2$. The etendue of the fiber is $(0.22 \text{ rad})^2*(200\times10_{-6} \text{ m})^2$, which is much less. Am I missing something here?

We believe the reviewer mistakenly used the full angle instead of the half angle when deriving the acceptance angle of the fiber. Please see our reply to referee comment #1.

If the statement by the reviewer were correct and the fiber were to collect light from a disc with 265 mm diameter instead of the entire full 500 mm diameter mirror, the discrepancy between the simulated and measured lidar return signal should differ by a factor of 4 (the collected signal is proportional to the area and thus the diameter squared). However, we found a discrepancy of only 38 %. This clearly indicates that the fiber collects light from a disc much larger than 265 mm in diameter.

Could the author also please explain why three detectors (two APDs and one PMT) are set up in the receiver?

Please see our reply to referee comment #1.

We added following text:

While the PMC backscatter signal can be handled by a single detector, we chose to add two additional detectors to cover the large dynamic range of the Rayleigh signal originating in the lower atmosphere closer to the gondola.

Minor comments:

Line 20, please consider to replace "wanted" with "desired"
Done.

Line 23, please consider to replace the reference Wickwar et al., 2016 with Sox et al., 2018 (a JRG paper for the same topic).
Done. Thank you for suggesting this reference.

Line 27, please consider to add "Krueger et al., 2015" as the reference on Na Doppler lidar.
Thanks for pointing us to this reference. We added it at the end of the sentence.

Line 35, One of the other advantage of the balloon lidar is that The high altitude flying lidar also avoids the scarified laser power and return signal due to strong scattering in the troposphere, suffered by the ground-based lidar system.
Yes, we agree. We added:
Additionally, such a lidar gains about 20 % in signal due to avoiding the significant optical extinction caused by scattering in the lower atmosphere.

Line 129, please consider to replace "Not converted" with "The residual".
Done.

Line 258, please consider to change the phrase "worst case estimated" to "worst case scenario estimation".
Done.

Line 262-263, the last sentence of the paragraph "Therefore, we defer the discussion to the discussion" sounds a little strange, please consider to rephrase.
We agree and changed this sentence to:
We will discuss these factors in Section 5.

Line 277-278, please explain the underline motivation to choose these two factors, 38% and 280%
When all known factors (e.g. laser power, telescope aperture, transmittance of filters) were taken into account, our simulations produced a simulated lidar signal which was larger than what our instrument detected in flight. However, when we decreased the simulated signal by 38 %, simulation and actual measurements were in good agreement. This let us conclude that we have unknown losses somewhere in the optical path, most likely due to a misalignment of the last lens in front of the detector. These losses amount to 38 % of the total signal, as estimated from the comparison between simulated and measured signal. Likewise we simulated the solar background which is supposedly detected by our instrument and compared those simulations with our measurements. It turned out that we had to increase the simulated background by 280 % in order to bring it into agreement with the measurements. Our interpretation is that additional scattered sun light was collected by the BOLIDE instrument via a process not included in our simulations. A likely candidate is scattering of diffuse sun light of the telescope spider.

We added following text:

With all known factors taken into account, our simulations yielded a simulated lidar return signal which is larger than what our instrument detected in flight. However, when the simulated signal is decreased by 38 %, simulation and actual measurements are in good agreement. This let us conclude that there are unknown signal losses somewhere in the optical path, most likely caused by misalignment of the last lens in front of the detector. On the other hand, the comparison between simulated background and measured background showed that our simulation is underestimating the background by 100 % to 300 % depending on time of day.

Line 324, please consider to delete "due to
Done. Thank you.

**Reviewer #2**

The manuscript decribes the design, development and first results of a novel lidar op-erated on a NASA long duration balloon mission. The lidar operated for 6 days in July2018 and sucessfully recorded Polar Mesospheric Clouds (PMC). The manuscript de-scribes the technical aspects of the lidar development and performance estimations forthe detection of PMC. The estimates are compared to one example on July 11, 2018.The paper is well written and allows for an easy understanding of the complicated topic.It was a pleasure to read the manuscript.

We appreciate the assessment by the reviewer.

I only have some minor comments in addition to those by Referee #1:

Page 4, Line 101: "180µrad"→"165µrad"?
Corrected. Thank you for spotting this inconsistent value.

Page 6, Line 136: "First mirror"→"Motorized mirror"
Done.

Page 6, Line 146: "100 Hz rotations..." what is the actual chopping frequency? For balance reasons I would guess the chopper blade has more than 1 hole.
We added the following sentence:
With three 60° long slits in the chopper blade, the resulting chopping frequency is 300 Hz.

Page 9, Line 197: what are "the laser replacement heater and radiator bias heaters"? Does that statement mean the same thing as the "replacement heaters" in line 191?

Please see our reply to referee comment #2.

We added the following text:
In order to maintain a constant heat load on the cooling system, laser replacement heaters, which dissipate the same amount of electrical power as the laser, are switched on whenever the laser is switched off. The radiator bias heaters are located on the radiator and are used

to control the temperature of the coolant flowing though the radiator by changing the temperature of the radiator surface.

 For me the detailed discussion of the plans for launches from Mc Murdo seem disturbing. At the end the summary is that 2∘to 45∘is the range of relevant solar elevation angles. Fig. 6 then shows solar elevation angles between<1∘and~40∘, I see no discrepancy here, but the text was just slightly confusing to read.

We added following text to the introduction starting at line 70 to clarify that all design work was done before the launch site was moved to the northern hemisphere:

Less than a year before the anticipated launch date it became clear that PMC Turbo would be unable to fly from McMurdo in the 2017-2018 austral summer, and NASA offered a launch from Esrage, Sweden (Fritts et al. 2019). Because BOLIDE was specifically designed for a flight in Antarctica and all design work was completed before the launch site was moved, in the following we will describe the original mission and mission requirements as much as possible and point to changes implemented in light of the flight in the northern hemisphere where necessary.

Page 12, Line 282: "We used this profile scaled..."→"We used the profile of$\beta_{PMC}$ scaled ..."
Changed.

Page 306 and 362: "...surpassed that of the largest ground-based lidar systems" and"...with better quality (higher resolution and lower detection threshold) than ground-based lidar experiments."

It is unclear what the actual parameters of the ground-based lidar systems are. I guess the statement is probably true at times, but I also guess that the performance of a ground based lidar varies substantially throughout the day, even more than the variation seen in BOLIDE (Fig. 6).

I suggest revisiting this statement, probably with updating the reference to "ground-based lidar systems" given on Page 3, line 79: There is one reference given to a rather old publication (Fiedler et al., 2011) where a decadal scale dataset was analized with a resolution that was for this purpose chosen to be 14 minutes. Since then a couple of publications have shown lidar observation of NLC with a resolution higher than 14minutes and some with resolutions well below 30 seconds and 100 m. Measurement uncertainties have been reported as low as $0.1×10^{-10}$ $m^{-1}$ $sr^{-1}$.Kaifler et al., JGR, 2018; https://doi.org/10.1029/

Ridder et al., JASTP, 2017; http://dx.doi.org/10.1016/j.jastp.2017.04.005
Fritts et al., JASTP, 2017; http://dx.doi.org/10.1016/j.jastp.2016.11.006
Kaifler et al., ACP, 2013; http://dx.doi.org/10.5194/acp-13-11757-2013
Baumgarten, GRL, 2012; http://dx.doi.org/10.1029/2011GL049935

Thank you for pointing us to above references. Ridder et al. state a resolution of 30 s and 40 m, but no information on the detection threshold is given. This is the same resolution as used in Kaifler et al. 2013, also without stating the detection threshold. Baumgarten et al. 2012 used data at 1 minute and 475 m resolution, but again no detection threshold is provided. Only Kaifler et al. 2018 provide clear information on vertical and horizontal resolution and detection threshold.

We added following text starting at line 87:

Kaifler et al. (2013) processed data acquired with the same instrument at 30 s and 40 m resolution, but no information on the detection threshold is provided.

We added following text starting at line 326:
Kaifler et al. 2018 achieved a higher peak sensitivity of $\beta = 0.1 \times 10^{-10} \text{m}^{-1}\text{sr}^{-1}$ with their ground-based system, but their integration time of 120 s is four times larger compared to BOLIDE and measurements were obtained for a few hours in darkness only, while BOLIDE acquired continuous PMC backscatter measurements with approximately constant sensitivity for days. We note that Kaifler et al. (2013) and Ridder et al. (2017) present PMC backscatter profiles at higher resolution of 30 s and 40 m, but no information on the detection threshold is given. We want to highlight that BOLIDE data can be processed at even high resolutions. Examples with a resolution of 10 s and 20 m are presented in Fritts et al. 2020.

Line 387:
We replaced
"It met all design requirements and provided science data of PMCs with better quality than ground-based instruments."
with
"It met all design requirements and provided science data of PMCs with better quality than continuously operating ground-based instruments."

**Reviewer #3:**

This paper describes the BOLIDE balloon-borne lidar experiment for the detection of polar mesospheric clouds. The advantage of being on a balloon is a signal gain due to the shorter distance between the lidar and the clouds, the guarantee of always having clear sky conditions and the reduction of background light during the day. This lidar is built to be installed on a balloon pod with other instruments to fly in the Arctic or Antarctic. It flew for 6 days in the Arctic region during the PMC turbo campaign in July 2018. A troubling point is that the document only describes the project to fly the lidar from Mac Murdo in Antarctica but it flew from Kiruna in the Arctic. This change of flight location needs to be explained.

We added following text to the introduction:
Less than a year before the anticipated launch date it became clear that PMC Turbo would be unable to fly from McMurdo in the 2017-2018 austral summer, and NASA offered a launch from Esrage, Sweden (Fritts et al. 2019). Because BOLIDE was specifically designed for a flight in Antarctica and all design work was completed before the launch site was moved, in the following we will describe the original mission and mission requirements as much as possible and point to changes implemented in light of the flight in the northern hemisphere where necessary.

The document is more of a technical report describing the instrument and its performance than a scientific document. For example, the description of the lidar subsystems in section 3 is very long and rather technical.

I suggest that we limit ourselves to the essential elements of the document and move the detailed description to the appendix.

Because the technical description and the performance analysis is the heart of the paper, we prefer to keep the current structure and not move the description to the appendix.

There are almost no scientific results presented in the document.

We added a panel to Figure 7 showing the height-time section of PMC backscatter for a 4-hour period centered at the time of the example shown in Figure 7a. These observations show very bright and variable PMC with changing layer width and vertical movement. This hints at wave breaking and instabilities. A detailed study is in preparation.

Some results from BOLIDE lidar during the PMC-Turbo campaignwere presented in Fritts et al (2019) but it would be useful for readers to show someexamples of observations of polar mesospheric clouds and Rayleigh temperature

In addition to the new panel in Figure 7, we added a reference to the publication by Fritts et al. which presents the study of bore events and highlights PMC backscatter profiles with extremely high resolution (10 s and 20 m).

**Changes unrelated to reviewer comments**

We added a reference to a recently published paper describing the thermal control system (Baturkin et al. 2020) in line 204.

[revised manuscript text omitted]

---

## Author Response (AR2)

**"A technical description of the Balloon Lidar Experiment BOLIDE" by Kaifler et al.**

We thank the reviewers for her/his comment which is reproduced below.

Line 383:
The authors state:
It met all design requirements and provided science data of PMCs with better quality (higher resolution and lower detection threshold) than continuously operating ground-based lidar experiments.

However this contradicts their own statement in Line 325 and lacks supporting data:
We note that Kaifler et al. (2013) and Ridder et al. (2017) present PMC backscatter profiles at higher resolution of 30 s and 40 m, but no information on the detection threshold is given.

It should be corrected that Kaifler et al. (2013) showed and analyzed PMC backscatter profiles with a resolution of 0.3 s and 25 m. (See their figure 9)

**Authors' response**

Kaifler et al. (2013) show data at 0.33 s and 25 m resolution as stated by the reviewer. Initially, we did not refer to this resolution in our work because data at this resolution in Kaifler et al. (2013) are raw photon count data, whereas we defined the quality of our data set in terms of resolution **and** PMC brightness detection threshold. To demonstrate that the BOLIDE data is at least of comparable, if not better, quality as the ALOMAR RMR data presented in Kaifler et al. (2013) we added the following plot (Figure 8b) which shows photon count data at comparable resolution.

[Figure]

**Figure 8.** Photon counts of a 10-minute section of the data shown in Figure 7c with (a) $10\,\text{s}\times100\,\text{m}$ resolution and (b) $0.3\,\text{s}\times25\,\text{m}$ resolution.

Figure 9 in Kaifler et al. (2013) shows a maximum of 3 photon counts per bin. Of course, the maximum photon count depends on PMC brightness – the brighter the PMC the more photons per bin are detected by the lidar. Since Kaifler et al. (2013) don't provide information on the corresponding PMC brightness of the data shown in their Figure 9, strictly speaking we can't compare the performance of the two instruments based on the photon count data. However, it is probably safe to assume that Kaifler et al. (2013) selected an example with a bright PMC and good SNR, as we did in our Figure 8. So assuming that the PMC brightness is indeed roughly comparable, our data shows a factor of ~5 larger maximum photon counts per bin (3 versus 15-20). Therefore we conclude that BOLIDE has a higher sensitivity than the ALOMAR RMR lidar in Kaifler et al. (2013). In this sense our statement "it met all design requirements and provided science data of PMCs with better quality (higher resolution and lower detection threshold) than continuously operating ground-based lidar experiments" also holds for high resolution photon count data.

In our opinion there is no contradiction, as claimed by the reviewer. Ridder et al. (2017) show data at 30 s and 40 m resolution, but no information on the detection threshold is given. We clearly defined the quality as dependent on resolution **and detection threshold**:

> Lines 80-85: The lidar is to provide PMC backscatter profiles with higher resolution and similar PMC detection threshold as the ground-based ALOMAR RMR-lidar i.e. 30 s temporal and 100 m vertical resolution with a detection threshold of $\Delta\beta PMC = 1 \times 10^{-10} m^{-1} sr^{-1}$. For comparison, Fiedler et al. (2011) used 14 min, 150 m, and $\Delta\beta PMC = 1 \times 10^{-10} m^{-1} sr^{-1}$ for processing ALOMAR data.

With no detection threshold provided, the quality of the Ridder et al. (2017) data set can't be assessed as per our definition of quality. Consequently, the reviewer can't claim that the Ridder et al. data set is of better quality just because of the higher resolution, or conversely, that the BOLIDE data set is of lower quality because of the lower resolution. Based on the law of energy conservation, it is clear that if the resolution is increased, the number of photons per bin decreases, as does the signal-to-noise ratio. As a consequence, the detection threshold increases. Thus, the resolution alone does not tell you anything whether you are able to detect any PMC. In the extreme case you can have a very high resolution data set which is completely useless for science because of the overwhelming noise content. Without any detection threshold provided, we refrain from discussing the Ridder et al. (2017) data set in terms of data quality in our manuscript.

To our knowledge, the study by Fielder et al. (2011) is the most recent work which defines resolution and detection threshold for the ALOMAR RMR lidar. If the reviewer can provide any further references, we are happy to include those and discuss the data quality in relation to our data set.

We made following changes in the manuscript:

Line 327: added sentence
*A subset of the data in Kaifler et al. (2013) is shown at a resolution of 0.33 s×25 m.*

Line 329: added
*as compared to the standard resolution 30 s×100 m*

Line 330: added two paragraphs as well as new Figure 8 shown above
*Figure 8 demonstrates the high resolution capabilities of the BOLIDE instrument. While Figure 8a shows a 10-minute section in 10 s×100 m resolution, the temporal and vertical resolution are increased by a factor of 33 and 4, respectively, in Figure 8b. At this high resolution the fine structure of the PMC layer with multiple sharply bounded sublayers of few ten-meter thickness becomes visible. Even higher resolutions up to 0.1 s and 5 m are possible and may be used where PMCs are bright i.e. the signal is large.*
*The comparison of our high resolution data (Figure 8b) with Figure 9 in Kaifler et al. (2013) reveals that maximum photon counts detected by BOLIDE are a factor of ~5 larger. We note that raw photon counts are uncalibrated data and depend on PMC brightness, hence a direct comparison of the performance of the respective instruments is not possible. However, given that authors tend to showcase their best examples, it is probably safe to conclude that the BOLIDE instrument has a higher sensitivity than the ALOMAR RMR lidar.*

Line 394:

[revised manuscript text omitted]